# Assessments of Process Parameters on Cutting Force and Surface Roughness during Drilling of AA7075/TiB2 In Situ Composite

**DOI:** 10.3390/ma14071726

**Published:** 2021-03-31

**Authors:** S. Parasuraman, I. Elamvazuthi, G. Kanagaraj, Elango Natarajan, A. Pugazhenthi

**Affiliations:** 1School of Engineering, Monash University Malaysia, Jalan Lagoon Selatan, Bandar Sunway 46150, Malaysia; 2Department of Electrical & Electronic Engineering, University Teknologi Petronas, Seri Iskandar 32610, Malaysia; irraivan_elamvazuthi@utp.edu.my; 3Department of Mechatronics Engineering, Thiagarajar College of Engineering, Madurai 625015, Tamil Nadu, India; gkmech@tce.edu (G.K.); pumech@tce.edu (A.P.); 4Faculty of Engineering, UCSI University, Kuala Lumpur 56000, Malaysia; cad.elango.n@gmail.com

**Keywords:** drilling, aluminum, composites, in situ, machinability

## Abstract

Reinforced aluminum composites are the basic class of materials for aviation and transport industries. The machinability of these composites is still an issue due to the presence of hard fillers. The current research is aimed to investigate the drilling topographies of AA7075/TiB_2_ composites. The samples were prepared with 0, 3, 6, 9 and 12 wt.% of fillers and experiments were conducted by varying the cutting speed, feed, depth of cut and tool nose radius. The machining forces and surface topographies, the structure of the cutting tool and chip patterns were examined. The maximum cutting force was recorded upon increase in cutting speed because of thermal softening, loss of strength discontinuity and reduction of the built-up-edge. The increased plastic deformation with higher cutting speed resulted in the excess metal chip. In addition, the increase in cutting speed improved the surface roughness due to decrease in material movement. The cutting force was decreased upon high loading of TiB_2_ due to the deterioration of chips caused by fillers. Further introduction of TiB_2_ particles above 12 wt.% weakened the composite; however, due to the impact of the microcutting action of the fillers, the surface roughness was improved.

## 1. Introduction

With the rapid development in the avionics industry, aluminum matrix composites (AMCs) are increasingly used for many features and reasons, such as enhanced mechanical and physical properties, better quality strength-to-weight ratio, improved ductility, ultimate values of modulus and strength, low coefficients of thermal expansion, tremendous wear, corrosion and creep resistance [1]. In the current scenario, the stir casting method is being adopted in the production of aluminum alloy components for aerospace and automobile applications [2,3]. The in situ stir casting method has been a good option as a production method, as reinforced AMC composites do have good bonding, less wettability, proper distribution of fillers and better mechanical properties [4,5,6]. The aluminum matrix composites have been synthesized using various fillers, such as SiC, graphite, ZrB_2_, TiB_2_ and Al_2_O_3_. Yi et al. [7] recently introduced titanium diboride (TiB_2_) for the reinforcement of AMC that resulted in good physical and chemical properties due to the excellent bonding of aluminum alloy and fillers. Ramkumar et al. [8] reinforced AMC with 3 wt. % of TiB_2_ and 1 wt.% of graphite and stated that in situ hybrid composite resulted in enhanced machinability due to the self-lubrication properties of graphite particles. David and Dinakaran [9] synthesized AA7075/ZrB_2_ AMCs with 0, 3, 6, 9 and 12 wt.% of fillers and reported that ZrB_2_ improved the properties of the resultant composite. Preetkanwal [4] reported in their review article that polycrystalline and diamond treated hardened steel tools are appropriate for high-speed turning and machining. In addition, at low feed and depth of cut, the surface finish was improved. Pandiyarajan et al. [10] reviewed and reported the effects of single and multi-reinforcement of AMC. Harpal Singh et al. [11] reveals from his study that addition of hard TiB_2_ particles improved the mechanical properties and tribological of composites. Optical photomicrographs showed microstructure that revealed uniform diffusion of ZrB_2_, either in spherical or elliptical shape. Properties such as tensile, wear and hardness of reinforced AMCs have been found and reported in the literature [12,13,14].

High surface finish is the requirement in machining of components. The variables such as cutting force, tool angle, depth of cut, speed and feed are independent parameters that control the surface roughness of the machining [15,16,17]. Some of the literature related to characterization of turning of AMCs is available. Anil et al. [18] produced graphite reinforced AMC and showed the effect of turning and drilling parameters on surface finish of the component. They also indicated that cutting force decreases upon increase in weight percentage of the filler. The decrease in cutting force helps in getting better surface finish. Tomadi et al. [19] synthesized Alsi/AlN composite and studied the optimum cutting parameters using Taguchi orthogonal array and signal-to-noise ratio. Ramkumar et al. [20] studied the problem pertaining to efficiency of turning cycles by measuring the cutting forces and tool wear on the cutting edge. Jimmy et al. [21] synthesized Al-Si-TiB2 composites and analyzed the performance characteristics of cutting forces and surface roughness. Basavarajappa et al. [22] synthesized 15 wt.% reinforced Al 2219 composite and 15–3 wt.% graphite reinforced A1 2219 hybrid composite. They studied the surface roughness and chips obtained from turning operations and concluded that the lowest surface roughness was recorded by the Polycrystalline diamond cutting tool (PCD) tool; moreover, surface roughness was lower for the coated carbide tool than the carbide tool.

The related literature in recent years is discussed below. Ismail et al [23] investigated tool wear, machining characteristics and chip formation using finite element analysis (FEA). They used the Johnson–Cook constitutive and fracture model and provided the guidance for controlling chip formation. Drilled Al6061/A1_2_O_3_ AMC was investigated [24] and the effect of percentage weight of alumina fillers, drill bit point angle, feed, speed and surface finish were demonstrated with results. Sultane et al. [25] studied the impact of uncoated carbide drill bit on hole quality and tool wear. Singh et al. [26] studied the dry sliding wear behaviour of AA6082–T6/TiB_2_ in situ composites using response surface methodology. Yasir et al. [27] studied SiC and alumina reinforced Al 6061 fusion composite and reported that the wear resistance of 15 wt.% filler reinforced hybrid composite is better than that of 5 wt.% filler reinforced composite. Zhu et al. [28] conducted a drilling of aluminum 2024-T351/titanium Ti6Al4V stack and studied the mechanism of chip formation. Krischner et al. [29] conducted drilling on Al/SiG/Graphite hybrid composite and reported influencing factors on surface roughness. Senthilbabu et al. [30] studied and revealed the formation of continuous chip during drilling and proposed a solution for the disposal of chips. They also analyzed the chip removing forces and motion of the chips during the drilling. The machining of the aluminum metal matrix (Al/SiC/20p) reinforced with silicon carbide particles through polycrystalline diamond (PCD) inserts was investigated by Palanikumar et al. [31]. Rajmohan and Palanikumar [32,33] explored the optimization of the drilling parameters of Al 356 aluminum alloy with silicon carbide and mica of 25 and 45 microns, respectively. A detailed study on tool wear during machining process was conducted by Ismail et al. [34]. They focused on drilling parameters, tool materials, different types of geometries, drilling environments, mechanism of tool wear and preventive measures against drill wear.

The following observations were made from this literature review. First, most of the research has been focused on turning and many other grades of aluminum composite. Little or no research has been done on reinforced AA7075 composite, and particularly on the assessment of drilling surface quality of the chosen composite. Therefore, the current research is aimed to assess the drilling features of in situ formed TiB_2_ reinforced AA7075 AMC. To achieve this goal, AMCs were synthesized through an in situ stir casting method with a varying percentage (0, 3, 6, 9 and 12 wt.%) of TiB_2_ fillers. The drilling of rectangular plates was conducted with a high-speed steel (HSS) drill bit under varying cutting speed and feed rate conditions. The cutting force and surface roughness of the machined sample were recorded and analyzed in terms of weight percentage of TiB_2_ fillers. In addition, the tool layer, chips, and machined surface morphology microscopic and macroscopic characteristics were also studied.

## 2. Material and Methods

In order to prepare the samples for drilling, aluminum matrix (AA7075) was purchased from Bharat Aerospace Metals., Mumbai, India. Titanium diboride (TiB_2_) is a ceramic powder with high hardness, high electrical and chemical properties. It is superior in corrosive resistance, which makes the material attractive for use in corrosive applications. The TiB_2_ reinforced aluminum metal matrix composite (AMC) is a good choice for aerospace and automotive parts. AMCs were synthesized through in situ stir casting method with varying weight percentage (0, 3, 6, 9 and 12 wt.%) of TiB_2_ fillers, as shown in Figure 1. First, Al alloy in short rod forms was melted in the furnace. The respective wt.% of TiB_2_ fillers were added into the molten Al alloy at 850 °C. Inorganic salts such as K_2_TiF_6_ and KBF_4_ were also added into the molten AMC. After thorough stirring of the molten AMC, the melted composites were cast into rectangular plate form and cylindrical form. The dimension of cylindrical samples was 45 mm in diameter and 450 mm in length. The rectangular samples were 100 mm × 100 mm × 100 mm. The hardness of samples follows: AA7075/ 0 wt.% TiB_2_ was 65 VHN, AA7075/ 3 wt.% TiB_2_ was 80 VHN, AA7075/ 6 wt.% TiB_2_ was 106 VHN, AA7075/ 9 wt.% TiB_2_ was 124 VHN and AA7075/ 12 wt.% TiB_2_ was 175 VHN.

Experimental studies were carried out in a VMC100 CNC milling machine (Perfection Global LLC, IL 60007 USA) by varying the cutting speed and feed rate under dry conditions. High speed steel (HSS) drill tools 6 mm in diameter and 118 degree point angle (HSS DIN338 standard) were used for conducting drilling experiments. The thrust force was measured using a three-dimensional dynamometer (Kistler 9257B, Singapore) with data acquisition system. The proposed machining parameters were a cutting speed of 1000 to 2000 m/minute in steps of 500 m/minute and a feed rate of 0.05, 0.10 and 0.15 mm/rev, with a constant cutting depth of 8 mm. No chip breaker was employed during the experiments.

The machined specimen and the collection of the respective chips during the experiments were carefully numbered and reserved for further characterization. The surface roughness and profile were measured and analyzed using a Surtonic S-128 roughness tester (Taylor Hobson, supplied in, Bangalore, India) in compliance with ISO4287 and the mean of surface roughness of five samples in each wt.% was recorded. Figure 1b shows a batch of AA7075/ TiB_2_ in situ casting plates before and after drilling. The samples of 10 × 10 × 5 mm^3^ in each weight percentage were taken for field-emission scanning electron microscopic (FESEM)(Hitachi High-Tech India Private Limited, Maharastra, India) examination. First, all specimens were polished using emery sheets (No. 600, 800, 1000, 1200, 1500) for an hour. Specimens were carved utilizing ordinary Keller’s reagent and then examined with the assistance of a discharge electron magnifying lens (Hitachi High-Tech India Private Limited, Haryana, India).

## 3. Results and Discussions

Figure 2 displays 50SE and 100SE magnifications of field-emission scanning electron micrograph (FESEM) images of AA7075/ 12 wt.% TiB_2_ AMC. A uniform scattering of TiB_2_ particles is observed at low magnification, as shown in Figure 2a. On high magnification, as shown in Figure 2b, a certain repetitive spider weblike pattern of TiB_2_ dispersion is observed. It is a typical feature of composites in situ [8]. There are a few TiB_2_ particles between the edges of the grain and within the grains. It is known that a segregated structure is generated by the impact of TiB_2_ particles. These dispersions could be responsible for the solidification pattern. Typically, due to the exothermic chemical reaction, the local surrounding temperature raises the fixed furnace temperature. The heat released during the reaction cycle increases the solidification time and minimizes the rate of cooling. It is observed from microscopic study that the front solidification forcibly shifted the dissolved particles to the position of the grain boundary. Particle trapping at an advance solidification front with interfacial curvature occurred. This lower interfacial effect between TiB_2_ particles and aluminum matrix could have allowed TiB_2_ particles to be rejected within the boundaries of the grain. Therefore, it appears like an Al grain island swallowed by particles from TiB_2_. Such micrograph tests were intended for the researchers to produce the composite successfully. The measurable elemental analysis of in situ AA7075/ 6 wt.% TiB_2_ composite was conducted using quantitative electron microprobe analysis by means of a lithium drifted silicon detector (energy dispersive X-ray spectroscopy method (EDAX) Hitachi S-300 Hitachi High-Tech India Private Limited, Haryana, India) using an operating voltage of 20 kv and 500× magnification.

Figure 3 and Table 1 depict EDAX spectrum and elemental composition of AA7075/ 6 wt.% TiB_2_ composite, respectively. It is observed from Table 1 that elements such as aluminum, copper, and titanium boride were existing in the sample. The other smaller crests depict the existence of Zn in the resultant composite. The optical microscopic image as shown in Figure 4 provide evidence for the particle refining action of TiB_2_ particles in AA7075/ 9 wt.% TiB_2_ in situ composite. The reason for the decrease in the size of grains is the rise of TiB_2_ particles. The refinement of grains could have taken place due to the following reasons.

The solidification process refrained from the face growth of in situ formed a-A1 particles. The TiB_2_ particles were solidified and formed as a nucleus due to aluminum grains. The entire microstructure shown in the optical micrographs (OM) could be considered as an island consisting of a high region of aluminum matrix surrounded by TiB_2_ particles, as evidenced in Figure 4. Due to the constitutional cooling zone before the particle, more than one nucleus per cell was produced while TiB_2_ particles were increased. The increase in TiB_2_ particles provided advanced resistance to aluminum matrix grain growth, which in turn resulted in grain refinement.

The electron backscatter diffraction (EBSD) maps shown in Figure 5a–e evidence the microstructure refinement and presence of the grainy structure. It is observed that TiB_2_ particles granulated as a refined grain in the composite and the impact of TiB_2_ particulate on the content of normal grain size is evidenced. Further observed is that the size of the grain was additionally refined as TiB_2_ particulates increased. There is no linearity in reduction of grain size and increase in the content of TiB_2_ particulates. This is due to a kinetic reaction that caused the steady state and minimized the grain size with increasing TiB_2_ (0, 3, 6, 9 and 12 wt.%). A coarsening particulate was noticed due to high reaction rate. The formation of different sized TiB_2_ particulates occurs because of constant reaction time, and this was the reason for the variation in the slope shown in Figure 5f.

In order to assess the behavior of the composite during the drilling process, it is essential to evaluate the effect of cutting forces on the speed. Figure 6a–c illustrates the influence of drilling speed verses cutting forces by varying feed rate levels of 0.5, 1.0 and 1.5 mm/rev.

Figure 6 is understood that the cutting force reduces with the increase of the drilling speed. These changes are due to the coefficient of thermal expansion of the TiB_2_ particles into aluminum matrix, which caused an immense dislocation in the composite material. These changes are also vulnerable to heat intensity or temperature. Indeed, the influence of the cutting speed may impact the heat generation and increased the magnitude of heat on the drilling surface. This increase induced the portion of dislocation and caused thermal distress and hence the lower cutting force was suggested to use at higher cutting speed. Considering the situation, due to particle displacement at low cutting speed and high strength of the composite there is a need for additional power. In addition, as cutting speed is increased, the contact distance between the chip and the drill tool is reduced and the propensity to remove the chip from the tool is increased. It was also found that, at lower cutting speed, the extracted metal chips may stick to the cutting tool which produced the built-up edge (BUE). The BUE formation was reduced when the cutting speed was increased. It is concluded from the observations that less cutting force is suffice for the drilling of the composite, when increasing the cutting speed of the tool.

The impact on the cutting force on TiB_2_ particles is depicted in Figure 7a–c. Adding the TiB_2_ particles into aluminum composites minimized the cutting force at the feed rate levels of 0.5 mm/rev, 1.0 mm/rev and 1.5 mm/rev. This could be due to dislocations generated at the time of in situ of the composite. The inclusion of TiB_2_ particles increased the density of the dislocation; in turn, it increased the hardness of the composite. This is a new observation noticed in the current investigation. The presence of reinforced particle caused the change in mechanism during drilling of the composite. Different patterns of composite deformation and chip formation are observed due to the nature of huge plastic deformation in shear. This particle reinforcement resulted as a broken chips formation or chip breaker and reduced continuous chip creation. TiB_2_ particles also decreased chip adherence with the cutting tool and reduced BUE height. Therefore, there is a lower cutting force as TiB_2_ particles increase. It is more apparent from the graphs that when speed increases from 1000 m/min to 2000 m/min, the cutting force is reduced. It is concluded that, rather than cutting pace, the influence of the TiB_2_ is protruding the cutting power.

Figure 8 shows the effect of the drilling speed over surface roughness. Interestingly, notice that the surface roughness decreases when cutting speed is increased. Due to the BUE and associated discontinuous chip formation during drilling, a smooth surface finish was achieved at a higher cutting speed. It was also observed from the drilled samples that they have deposited BUE layer. Two kinds of discontinuous chips, viz., spiral shape and string shape, were observed during the drilling of AMCs at lower cutting speed (1000 m/minute). Moreover, spiral chips were in a greater amount and string chips were in less number.

The influence of TiB_2_ in aluminum matrix and the effect on surface roughness with different drill speeds are shown in Figure 9. Variations are noticed in the distortion pattern of aluminum composites with TiB_2_ particles. It is obvious that aluminum matrix deforms quite easier than TiB_2_ particles. The difference in the deformation of base matrix and the filler could have caused TiB_2_ particles to get fracture during the drilling. The shattered particles split the machined layer randomly at the microlevel. The chips were broken by the impact of the reinforced particles. All fractured and discontinuous microchips ensured the high surface roughness. The increase in TiB_2_ fillers in composite made the drilled surface even rougher.

Comparing the research findings of Zhaoju et al. [28], the present study proves that the best filler improves the surface roughness. Figure 10a, b exhibits the morphology of BUE formed in a flank surface of the tool during drilling of AA7075/ 0 wt.% TiB_2_ AMC at drilling speed of 1000 m/minute. The temperature and friction at the cutting point was observed to be higher, which caused the chips to join locally together at the tool surface. The rise in temperature caused the successive layer to adhere into the previous layer. This is a reason for observing a vivid BUE in the flank face of the cutting tool. BUE subsequently made a way for the change in flash angle and the direction of the chip. As a result of this, an increase in force was required for continuing the drilling, which is also evidenced and discussed. The BUE formed during the drilling caused an irregular pattern of deformation and stress. Whenever the height of BUE exceeded a certain point, due to shear stress failure it was removed from the flank head. All formation and separation were replicated cyclically during drilling.

Figure 10b exhibits the tip morphology of the tool during drilling at 1000 m/minute. For the higher content of TiB_2_, the wear was found to be higher in the lip edge. It was observed that increased titanium diboride particles improved the abrasive action of the tool edge and hence wear on the cutting edge was enhanced. The hacksaw tooth shape was formed on both the lip faces and cutting edges, although the height of the hacksaw tooth shape was greater in the cutting edge and, moreover, it decreased towards both lip faces. This tendency of the hacksaw tooth inflicts the damage to the drilled surface. Moreover, the hacksaw tooth shape could break the chips into cone and triangular shapes at higher drilling speed and content of TiB_2_ particles. It is concluded from the observations that the weight percentage of TiB_2_ particles makes a significant contribution in dictating the tool life.

Another typical indicator for evaluating tool geometry is the chip morphology produced during the drilling process. Figure 11a–j shows the photographs of chips created during drilling of AA7075/ 0, 3, 6, 9 and 12 wt.% TiB_2_ AMCs under extreme cutting conditions. It is understood from the results that no continuous chip was formed, and the semicontinuous chips were seen due to their disposable nature. On comparing the formation of chips during drilling with the conventional cutting process, three salient points are noted. First, there was no breakup of chips when the chips were in contact with the cutting edge. There was further deformation of chips due to drill flute-land and hole-wall. Second, the direction of the chip flow was regulated due to the difference in drilling speed along the cutting edge. The initial chips were found to be cone shaped, as the cutting speed was less at the point closer to the inner cutting edge than to the peripheral of the drill. The chips tried to move towards the drill center. Third, the interaction of chip and drill flute is a function of the geometry of the drill point or point angle and geometry of the drill flute, or, in other words, the helix angle. The edge of the point angle and helix angle caused the change in deformation that led to the change in the shape of the chips. Although chips were initially spiral cones, they were changed to several other shapes while the drill tool penetrated deeper. The shape of the chip is one of the main factors for a smooth drilling process. The breaking of chips during the drilling also helps in conducting a smooth drilling.

The drilling of Al/TiB_2_ composite resulted in spiral and string chips due to the ductile property of the material, but it resulted in cone and triangular shapes when there was an increase in weight fraction of TiB_2_ in the composite. The thickness of the chips was increased when the feed was increased. The concurrent increase in weight percentage of TiB_2_ and feed resulted in one or two spiral and string chips, followed by a greater number of cone and triangles chips. Once it started resulting in chips, the inner cutting edge went in at a significant manner slower than the outer cutting edge, and, moreover, the inner cutting edge became shorter than the outer chip. The change in chip size compelled its flow into the center of drill rather than being perpendicular to the cutting edge.

Figure 12a,b depict the chips’ back surface morphology. Figure 13a displays the plastically deformed surface of the chip, which resulted from AMC with 3 wt.% of TiB_2_ drilled at the cutting speed of 1000 mm/min. The low amount of TiB_2_ caused the composite to deform very quickly due to the temperature increase during drilling. As a result, a coarse saw tooth appearance was noticed.

Figure 13b shows the morphology of the chip that resulted from the same sample drilled at a cutting speed of 2000 mm/minute. The significant influence of the cutting speed is observed from the morphology of the chips. The plastic deformation resulted in a higher temperature at a lower cutting speed. As temperature rises, the aluminum matrix becomes plasticized and plastically deformed. Hence, the length of the chip was observed to be smaller for higher cutting speed. In addition, Figure 13b displays the segmented surface having parallel sections. Withstanding plastic deformation of aluminum matrix was favored with an increase in TiB_2_.

Figure 13a reveals the FESEM morphology of AA7075/ 0 wt.% TiB_2_ AMC at cutting speed 1500 m/minute and feed rate of 0.05 mm/rev. A moderate softening of aluminum matrix was observed, along with fewer feed marks and symptoms regarding materials deposited on the drill flankface. The heat energy generated at the cutting speed was less compared to the cutting speed of 2000 m/minute.

The lower cutting speed, in other words, maximized the residual time of the heat produced per unit drilling distance. The heat produced during the drilling process softened the aluminum matrix. The low cutting speed created a kind of ridge across the drilled surface of the feed mark. The tool BUE attempted to get away during drilling and the defected material was flown to adhere to the surface of the material. These sorts of conditions created a deprived drilled surface and enhanced surfaces through smooth roughness. The machined surface at a high cutting speed showed parallel feed mark ridges. It was also observed that the edges of the feed mark were found to be not sharp, but still exhibited a blended softened surface. Hence, the surface roughness was observed to be lower for higher cutting speed.

Figure 13b describes the morphologies for the drilled surface at a speed of 1000 m/minute and feed rate of 0.15 mm/rev. There was no proof for the existence of transported material from tool to either surface. The feedback ridges appeared to be defused for a higher content of TiB_2_ particle. In some places of the drilled surface, the ridges became blurred. These regions were characterized by way of the macrocutting matrix. From these observations, it was understood that filler particles were removed from the composite while drilling. The grainy edges of all these extracted bits made cutting inscriptions over the drilled surface. Hence, the surface roughness of the sample appeared to be higher for higher weight percentage of TiB_2_.

The surface finish of any drilled part is decided by the optimum drilling condition. The extensive analysis described above on drilling parameters and the impact of TiB_2_ filler particles is essential for selecting the optimum drilling parameters. In the application of medical implants, particularly for the spine implants, the highest surface finish is expected during the machining. Moreover, the implant must not lose its strength during drilling, turning and milling operations. In this point of view, these analyses will be helpful in designing medical implants. This study reports the advantages of reinforcement of appropriate TiB_2_ fillers into aluminum matrix and its effect on drilling parameters.

## 4. Conclusions

AA7075/ 0, 3, 6, 9 and 12 wt.% TiB_2_ AMCs were fabricated by an in situ stir casting method. Drilling experiments were conducted in a CNC vertical milling machine for various cutting speeds and feed rate under dry conditions. FESEM, optical micrographs (OM), EDAX and EBSD were used to study the morphology of the cutting tool, machined surface and chips. FESEM and OM showed almost all TiB_2_ particles were situated in intergranular regions. The microstructure of the composites exhibited a fair homogenous dispersion of TiB_2_ particulates in the sizes of nano, submicron and micron level, and various shapes. The EDAX patterns evidenced Ti peaks and confirmed the presence of fillers in the composite. EBSD maps of composites showed microstructure refinement and presence of the grainy structure. There was no linear relationship between grain size and TiB_2_ particulate, however, because of the variation in the size of particles and the rate of reaction. The cutting force was observed to be reduced as the cutting speed increased. This resulted from thermal softening and the reduction of dislocation density. Meanwhile, the cutting force increased as the feed rate increased. Due to large plastic shear, TiB_2_ particles acted as chip breakers and produced chip forming, which resulted in reduced cutting power. Interestingly, note that the surface roughness was reduced upon increasing the cutting speed. Meanwhile, it increased upon increase in weight fraction of the filler. The research analysis presented in this article will be helpful in selecting the optimum drilling condition for AA7075/TiB_2_ metal matrix composite.

## Figures and Tables

**Figure 1 materials-14-01726-f001:**
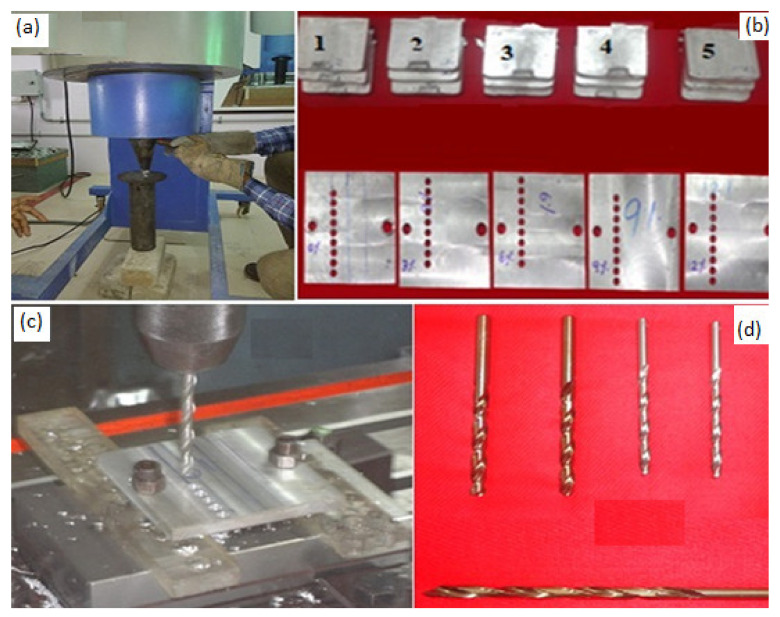
(**a**) Synthesis of in situ AA7075/TiB_2_ composites. (**b**) Samples before drilling and after drilling (0, 3, 6, 9 and 12 wt.%). (**c**) A sample at drilling and (**d**) the drill bits used.

**Figure 2 materials-14-01726-f002:**
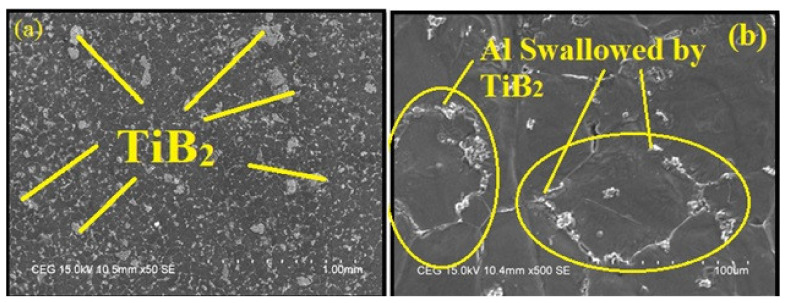
Field-emission scanning electron microscopic (FESEM) images of AA7075/12 wt.% TiB2 aluminum matrix composites (AMC) at (**a**) 50 SE (low magnification) and (**b**) 100 SE (high magnification).

**Figure 3 materials-14-01726-f003:**
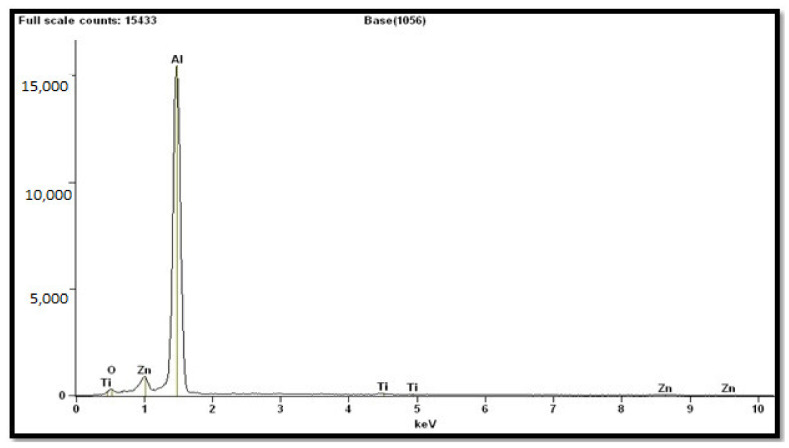
Quantitative electron microprobe analysis of AA7075/6 wt.% TiB_2_ in situ composite.

**Figure 4 materials-14-01726-f004:**
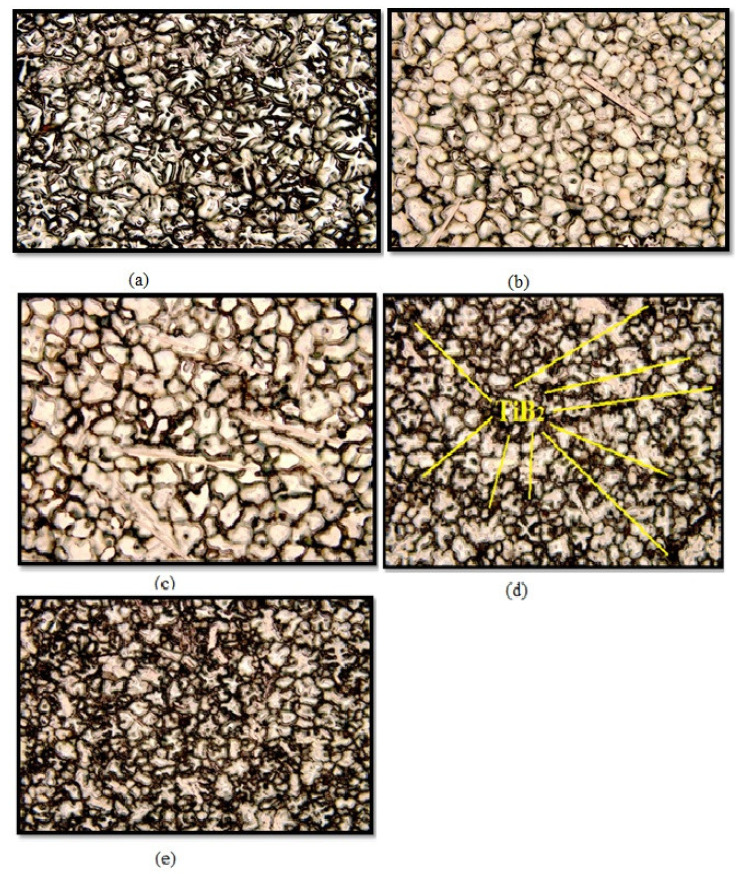
Optical microscopic image of AA7075/TiB_2_ composite (**a**) 0 wt.% TiB_2_, (**b**) 3 wt.% TiB_2_, (**c**) 6 wt.% TiB_2_, (**d**) 9 wt.% TiB_2_ and (**e**) 12 wt.% TiB_2_.

**Figure 5 materials-14-01726-f005:**
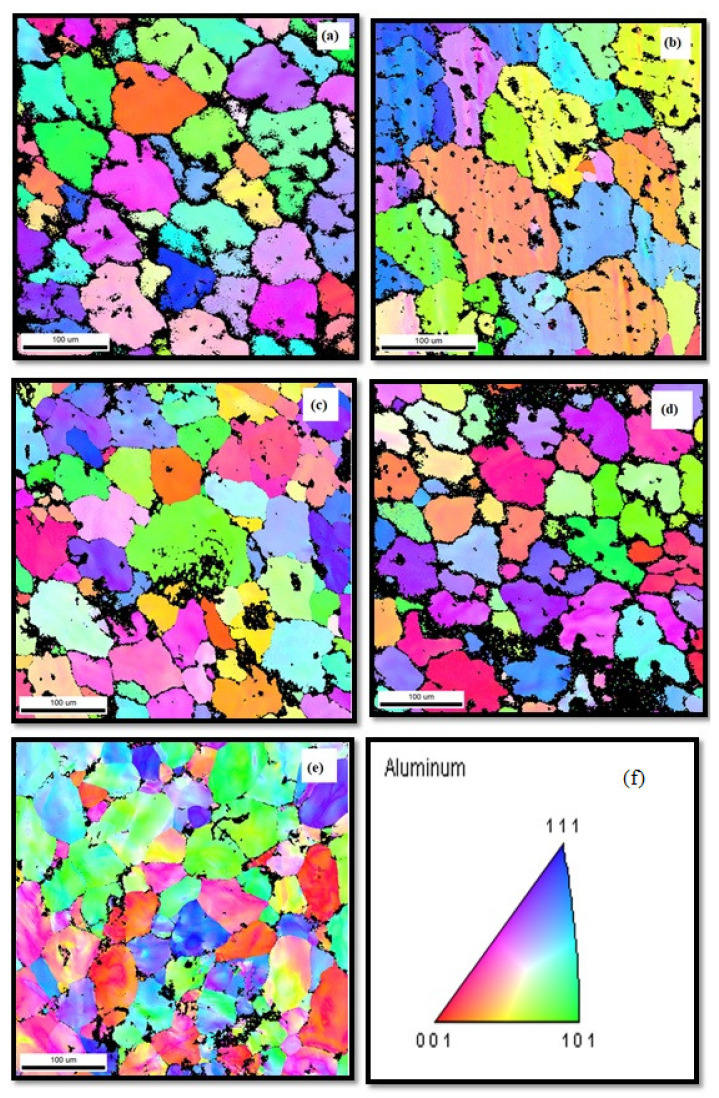
(**a**)–(**e**) Electron backscatter diffraction observations of (**a**) 0 wt.% TiB_2_, (**b**) 3 wt.% TiB_2_%, (**c**) 6 wt.% TiB_2_, (**d**) 9 wt.% TiB_2_ and (**e**) 12 wt.% TiB_2_. (**f**) slope variation.

**Figure 6 materials-14-01726-f006:**
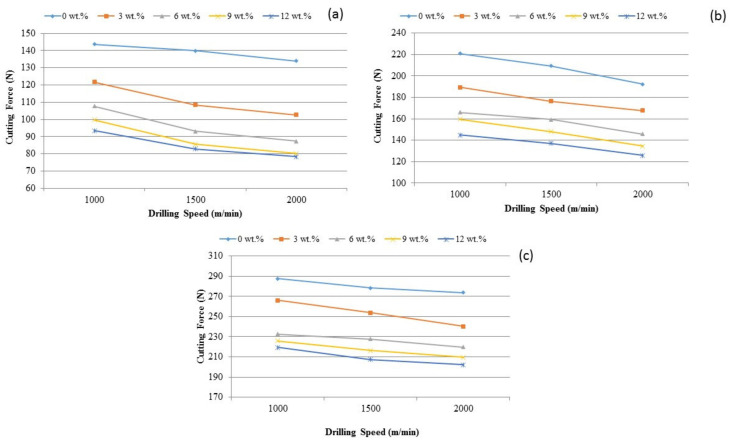
The influence of TiB_2_ on cutting force under various drilling spindle speed and feed rate levels such as (**a**) 0.5 mm/rev, (**b**) 1.0 mm/rev and (**c**) 1.5 mm/rev.

**Figure 7 materials-14-01726-f007:**
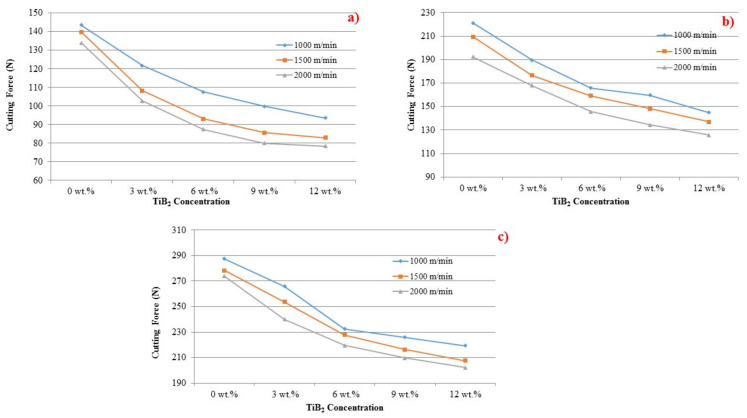
The effect of fillers on cutting force at various feed rate levels: (**a**) 0.5, (**b**) 1.0 and (**c**) 1.5 mm/rev.

**Figure 8 materials-14-01726-f008:**
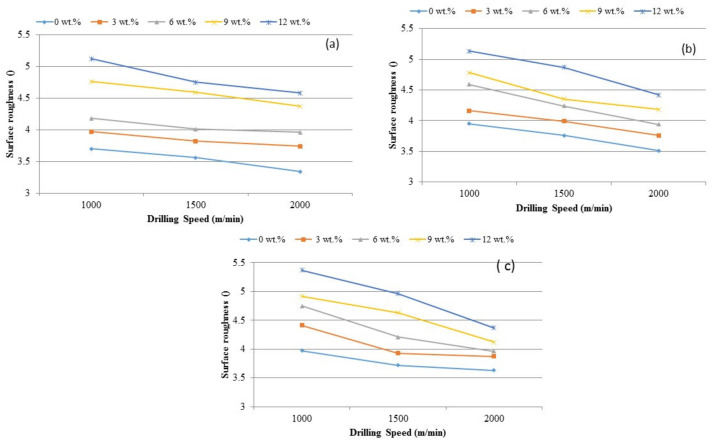
The influence of TiB_2_ on cutting force under various surface roughness and feed rate levels (**a**) 0.5 mm/rev, (**b**) 1.0 mm/rev and (**c**) 1.5 mm/rev.

**Figure 9 materials-14-01726-f009:**
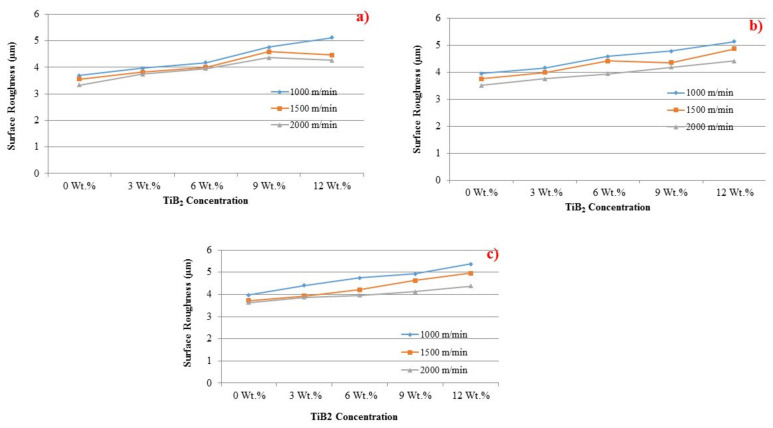
Influence of TiB_2_ particles on surface roughness at the feed rate levels of (**a**) 0. 5 mm/rev, (**b**) 1.0 mm/rev and (**c**) 1.5 mm/rev.

**Figure 10 materials-14-01726-f010:**
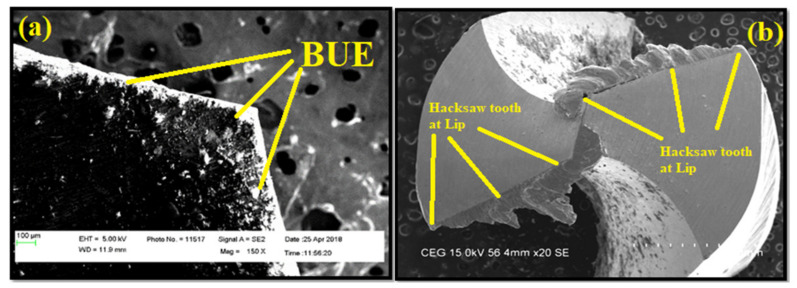
FESEM microscopic images of the (**a**) drill flank of AA7075, 0 wt.% of TiB_2_ AMC at 1000 m/minute; and (**b**) drill tip face AA7075, 3 wt.% TiB_2_ AMC at 1000 m/minute.

**Figure 11 materials-14-01726-f011:**
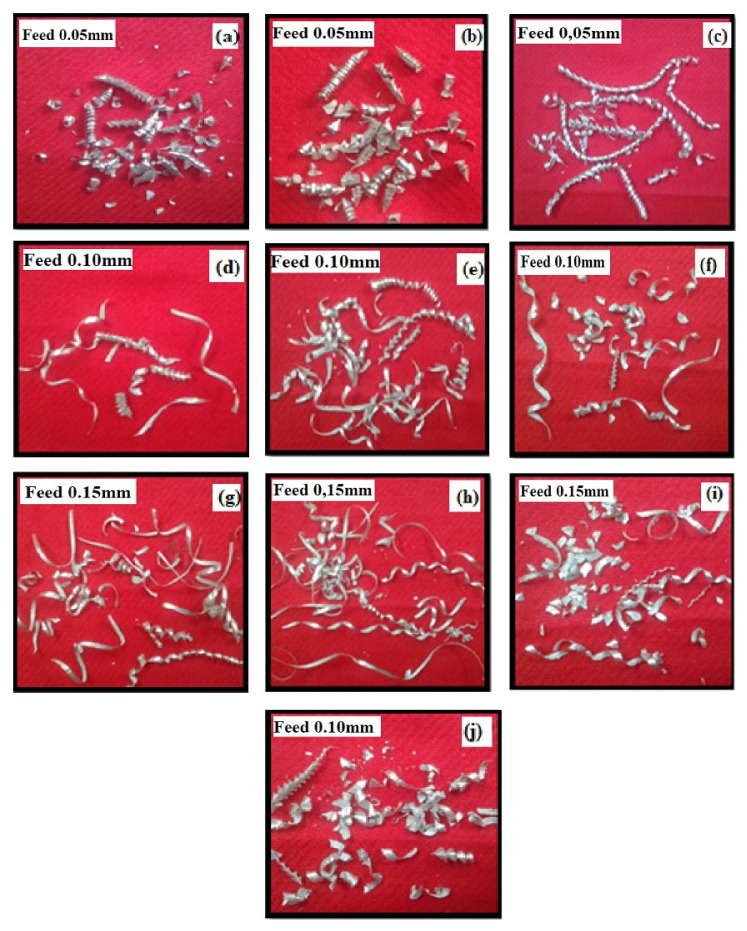
Photographic images of metal chips at (**a**) 0 wt.% TiB_2_ at 1000 m/minute, (**b**) 0 wt.% TiB_2_ at 1500 m/minute, (**c**) 3 wt.% TiB_2_ at 1000 m/minute, (**d**) 3 wt.% TiB_2_ at 2000 m/minute, (**e**) 6 wt.% TiB_2_ at 1500 m/minute, (**f**) 6.wt.% TiB_2_ at 2000 m/minute, (**g**) 9 wt.% TiB_2_ at 1000 m/minute, (**h**) 9 wt.% TiB_2_ at 1500 m/minute, (**i**) 12 wt.% TiB_2_ at 1000 m/minute and (**j**) 12 wt.% TiB_2_ at 1500 m/minute.

**Figure 12 materials-14-01726-f012:**
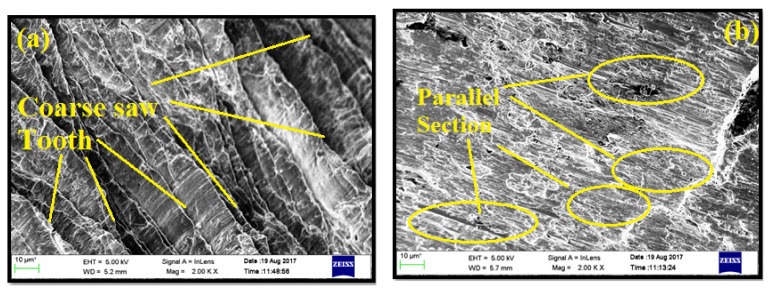
FESEM micrograph images of chip back surface in drilling: (**a**) AA7075, 3 wt.% TiB_2_ AMCs at 1000 m/minute; (**b**) AA7075, 6 wt.% TiB_2_ AMCs at 2000 m/minute.

**Figure 13 materials-14-01726-f013:**
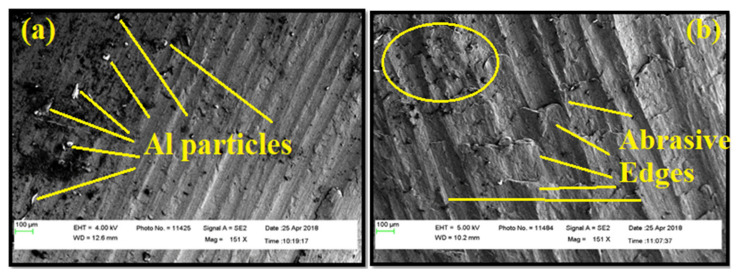
The morphologies for drilled surface: AA7075, 0 and 9 wt.% TiB_2_ AMC at cutting speeds of 1000 and 1500 m/minute and feed rates of 0.05 and 0.15 mm/rev (**a**) FESEM morphology of AA7075/ 0 wt.% TiB_2_ AMC at cutting speed 1500 m/minute and feed rate of 0.05 mm/rev and (**b**) morphologies for the drilled surface at a speed of 1000 m/minute and feed rate of 0.15 mm/rev.

**Table 1 materials-14-01726-t001:** Elements observed from quantitative electron microprobe (EDAX) analysis of AA7075/6 wt.% TiB_2_ composite.

Element	Net Counts	Weight%	Atom%
**O**	2047	4.33	7.17
**Al**	185119	93.06	91.50
**Ti**	956	1.79	0.99
**Ti**	20	-	-
**Zn**	79	0.82	0.33
**Zn**	8289	-	-
**Total**	-	100.00	100.00

## Data Availability

Data could be given upon request if anybody required.

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
