# Peer review of "Assessments of Process Parameters on Cutting Force and Surface Roughness during Drilling of AA7075/TiB2 In Situ Composite"

_materials, 2021, doi:10.3390/ma14071726_

Round 1

Reviewer 1 Report

line 135 „The melded TiB2 elements could have travelled over the aluminium matrix due to the influence of the frontend solidification.” I think this is not scientifically correct. There are no melted TiB2 elements, This is dissolved elements.

Figure 7. The figure is required to be understandable without the main text of the paper. The title of the figure: „The influence of drilling speed over cutting force at feed of (a) 0.05 (b) 0.10 (c) 0.15 mm” What is x mm? In the figure what is 0wt%, 3 wt%?

Please use consequently wt%, now sometimes Wt% sometimes wt%

Fig 12 not sharp enough.

The formation of the TiB2 is not written and explain in the paper, please add information about the formation of this compound.

Author Response

Review reply attached

Reviewer 2 Report

The reviewer comments of the paper «Assessments of Drilling Characteristics of AA7075/TiB2 in-situ Composite»- Reviewer

The authors presented an article «Assessments of Drilling Characteristics of AA7075/TiB2 in-situ Composite». However, there are several points in the article that require further explanation.

Comment 1:

Overall, the introduction is well written and understandable. However, it is helpful to add an article: Comprehensive Study on Tool Wear During Machining of Fiber-Reinforced Polymeric Composites. https://doi.org/10.1007/978-981-33-4153-1_5

For the purpose of the article, write more specifically what parameters will be investigated in the article.

Comment 2:

The quality of the all figures and their resolution should be greatly improved. Are all figures original? If not needed appropriate citations and permissions.

In the captions, you need to make an explanation, for example, in Figure 2, what do the values in% mean? In Figure 3, what does x50 and x100 mean? Add a scale bar for the figures 2, 3, 5, 11, 12. Sign for which drilling modes figures 3, 5, 6, 11, 13 were obtained.

What's the "feed"? Check this value and its units! mm/??

Comment 3:

Instead of a large photo of Machine 1, take a more local photo with the symbols. What and where? Show a photo of a drill with linear and angular dimensions. Give these parameters in the text of the article: diameter, angle at the tip, drill overhang, etc. What is the material of the cutting part? Describe all this in detail.

Give a diagram of the thrust force and roughness measurement. What is the hardness of the workpiece and how was it measured?

Besides the cutting speed Vc (m/min) it is also important to know the spindle speed n (rpm). The authors should revise all physical quantities in the article and, if necessary, replace them with standard ones. To do this, use reputable directories.

For devices and machines used in research, indicate in parentheses (manufacturer, city, country).

Comment 4:

It will be useful to add a section of Nomenclature in which to sign all the physical quantities and abbreviations encountered in the article. There are many physical quantities in the text and such a section will help to find the description of the necessary element.

For example,

n                : Spindle speed (rpm)

AMCs       : Aluminium matrix composites

etc.

Comment 5:

Conclusions. Draw quantitative and qualitative conclusions.

Use the format:

Conclusion 1

Conclusion 2

...

In addition, it is necessary to more clearly show the novelty of the article and the advantages of the proposed method. What is the difference from previous work in this area? Show practical relevance. Conclusions should reflect the purpose of the article.

Comment 6:

In the title of the article, it is preferable to briefly designate the parameters under study.

The article is interesting and helpful. However, in its current form and low quality and resolution of drawings, the article is not acceptable for the international journal. Authors should carefully study the comments and make improvements to the article step by step. Mark all changes in color. After major changes can an article be considered for publication in the "Materials".

Author Response

Review Reply attached

Round 2

Reviewer 1 Report

I think this paper is acceptable in this form

Reviewer 2 Report

The authors have improved the article according to the comments. However, some comments are not resolved.
1. Describe the measurement of the hardness of the workpiece. What hardness?
2. All figures are of very low quality and clarity and should definitely be improved. Now they look like raster very rough graphics. This is not valid for international journal.
After these shortcomings have been rectified, the article can be considered for publication.

Round 3

Reviewer 2 Report

The article can now be published.